trauma-informed care; organisational implementation; scoping review; low- and middle-income countries

**Corresponding author:**
Nicole Maiorano;
Email: maiorann@tcd.ie

# Trauma-informed care (TIC) in low- and middle-income countries: A scoping review of organisational implementation efforts

Nicole Maiorano[1,2] , Magdalena Wagner Manslau[1], Greg Sheaf[3], Mel Ó Súird[1,4], Tooba Nadeem Akhtar[1,4], Ruth Collins[4], Caoimhe Doyle[4], Doaa Sherif Karam[4], Caledonia Steltzner[4] and Meg Ryan[1,4]

[1]Centre for Global Health, Trinity College Dublin, Ireland; [2]School of Medicine, Trinity College Dublin, Ireland; [3]The Library, Trinity College Dublin, Ireland and [4]School of Psychology, Trinity College Dublin, Ireland

## Abstract

Due to an increased awareness of the prevalence and impact of trauma, "trauma-informed care" (TIC) was developed as an organisational framework aiming to centre the needs of survivors of trauma. TIC proposes that organisations can reduce trauma exposure by embedding specific principles (e.g., safety and trust) at every level of an organisation, improving the organisation for both service users and providers. Recent reviews of TIC implementation efforts have demonstrated its use in diverse settings; however, studies are overwhelmingly situated in high-income, predominantly English-speaking countries. Rather than reflecting a lack of TIC efforts in low- and middle-income countries (LMICs), these findings may be a result of the newness of the term TIC. To create a more inclusive evidence map, the current review captures efforts conducted in LMICs that may or may not use the label of TIC but align with the organisational approaches and key principles of TIC. A search of four databases and review of relevant references yielded 3,091 results, of which 255 met the inclusion criteria. Implementation efforts took place across 39 LMICs. The vast majority included involvement of another country, most commonly the United States. Approximately 90% of efforts were implemented within medical settings, and 69% focused on the TIC principle of cultural, historical and gender issues. The results of the current review have both theoretical and applied implications for TIC research. They query how and by whom TIC is conceptualised and defined, and how TIC aligns with other global research approaches. Results also highlight the need for organisational TIC interventions to conduct comprehensive baseline assessments of current efforts before implementing new efforts to avoid unintentional duplication. As the adoption of TIC frameworks becomes more widespread, it is imperative to increase research efforts aimed at developing a more thorough and inclusive definition of TIC.

## Impact statement

In response to global evidence of the prevalence and impact of psychological trauma, trauma-informed care (TIC) frameworks are growing in popularity. TIC provides an organisational framework to reduce the impact of trauma by preventing traumatisation and reducing re-traumatisation. Thus, TIC frameworks are proposed as a strategy to support service users' and providers' engagement with services and organisations. At this critical stage of development, recent reviews have aimed to capture organisational implementation efforts aimed at actioning. Although these reviews have captured a breadth of research, the efforts captured are almost exclusively implemented in high-income countries, creating TIC evidence maps that exclude low- and middle-income countries (LMICs). Due to variations in the experience of trauma and organisational structures across context and culture, this leads to models and understandings of TIC with limited generalisability.

The current review captures implementation efforts in LMICs that align with the organisational approach and principles proposed by TIC without utilising that term. In doing so, the review has both practical and theoretical implications. Practically, the review indicates a need for baseline organisational assessments that precede TIC implementation to capture efforts that are aligned with TIC, regardless of whether they label themselves as such. Theoretically, this raises questions about how researchers place TIC within Global research. As the model develops, researchers will need to identify the relationship between TIC and other related implementation efforts to support a more holistic understanding of how TIC fits into Global quality-improvement efforts.

## Introduction

Defining psychological trauma (hereafter referred to as "trauma") presents numerous challenges for the field of Psychology. The psychodynamic approach describes trauma as a response to event(s) that overwhelm one's ability to cope (Perrotta, 2019). Modern Western conceptualisations of trauma often refer to the diagnosis of post-traumatic stress disorder (PTSD), a diagnosis first recognised by the American Psychological Association (APA) in the 1980s (APA, 1980). Within the evolution of PTSD, there has been an ongoing debate about what *type of events* are considered traumatic (McNally, 2003). These differences in conceptualisations contribute to challenges in quantifying traumatic experiences. Utilising the Composite International Diagnostic Interview version 3.0 based on the Diagnostic and Statistical Manual of Mental Disorders-Fourth Edition's definition of PTSD, a global study analysed the occurrence of 27 traumatic events, including forms of collective violence, physical harm, interpersonal violence, interpersonal and sexual violence, injuries and other traumatic events. This study found that 70% of the general population experience at least one psychologically traumatic event in their lifetime, and 30.5% experience four or more traumatic events, indicating the widespread prevalence of such experiences. Although the authors noted that the definition of traumatic events had changed since the study was conducted, this study suggests that trauma experiences are relatively common globally (Benjet et al., 2015).

Decades of research link potentially traumatic events to a wide range of adverse outcomes, including poorer physical health (e.g., cardiovascular diseases and hypertension) and mental health (e.g., suicidality and depression; Stein et al., 2010; Husarewycz et al., 2014; Vibhakar et al., 2019). Due to the high prevalence of trauma within the general population, this has significant implications for public health and wider society (Watson, 2019). Individuals impacted by trauma are disproportionately represented within specific organisational settings – for example, health services – due to the relationship between trauma and poor health outcomes (Husarewycz et al., 2014). During treatment, additional traumatic events may be experienced by both service users and providers. For example, Nagle et al. (2022) found that 18% of patients in an Irish maternity hospital reported experiencing a psychologically traumatic birth, and the experience was significantly correlated with a history of depression and complications during birth. For hospital staff, experiencing critical incidents has been identified as a risk factor for negative mental health outcomes (e.g., PTSD, anxiety and depression; de Boer et al., 2011). These critical incidents may also contribute to high staff turnover within medical settings (McDermid et al., 2020). Collectively, this research indicates that traumatic experiences may not only affect service users and providers, but also the wider organisation.

Trauma-informed care (TIC) is a systematic approach that recognises the impacts that trauma has within organisations and seeks to reduce the likelihood of causing trauma or exacerbating existing trauma (i.e., re-traumatisation). First coined by Harris and Fallot (2001), numerous models have been developed for TIC implementation. The United States Substance Abuse and Mental Health Service Administration (SAMHSA) has been a notable thought leader for TIC, developing a range of theoretical and practical guides for its use (SAMHSA, 2023). SAMHSA proposes six core principles of TIC (safety; trustworthiness and transparency; collaboration and mutuality; empowerment, voice and choice; peer support; and cultural, historical and gender issues) that can be applied to all organisational components (e.g., financing and training). SAMHSA proposes that through implementing this framework, organisations can better support all individuals who interact with the organisation, including both service users and providers (SAMHSA, 2014).

Multiple literature reviews have been conducted to understand how TIC has been implemented. Although some reviews have focused on specific settings (e.g., child welfare; Zhang et al., 2021), three literature reviews (Bargeman et al., 2022; Mahon, 2022; Berring et al., 2024) and one umbrella review (Mahon, 2024) have aimed to capture the implementation of TIC more broadly. These reviews have used similar methodologies with a few distinctions in their search and screening approaches. Collectively, these reviews indicate a lack of consistency in TIC definitions and models. This may be affected by the newness of TIC. Berring et al. (2024)) highlighted the rise in the number of studies that met their inclusion criteria, particularly from 2015. The relatively recent development of TIC may limit the consensus of key TIC concepts and operationalisations. Mahon (2022) proposed that the newness of TIC research may also explain concerns in the quality of existing research. Mahon identified limits to the transferability of TIC findings due to the use of small, qualitative evaluations that are often conducted in one organisation. Further, although not highlighted as a finding within Mahon's (2024) umbrella review, the vast majority of the research included in the reviews is from high-income, Western, predominantly English-speaking countries. Overall, this raises concerns about the generalisability of existing TIC evidence.

The limitation of TIC research to Western, high-income countries (HICs) represents a significant gap in research due to the highly contextualised nature of trauma. Specifically, the experience of and response to trauma is subject to both cultural and contextual influences and varies across settings (Patel and Hall, 2021). Compared to HICs, low- and middle-income countries (LMICs) may be exposed to different potentially traumatic events at higher rates, such as forcible displacement (UNHCR, n.d.) or negative health effects of climate change (World Bank, 2024a). The way these experiences are interpreted also varies, with researchers challenging the application of a Western understanding of trauma rooted in diagnostic categories like PTSD to disparate settings (Marsella, 2010). This challenge extends to the use of Western treatments being applied to non-Western contexts. Honwana (2018) describes the harm that was caused by attempts from international organisations utilising talk therapy to treat child soldiers returning to Mozambique post-war. Honwana continues to describe the community responses in both Mozambique and Angola that were used to reintegrate the children into society based on societal values and beliefs. In a longitudinal study that followed child soldiers in Mozambique from 1988 to 2004, both the former child soldiers and members of their communities identified traditional ceremonies as supportive of the child soldiers' reintegration into society. The ceremonies were "local beliefs put into practice, they facilitated the realignment of individual, family and communal relationships" (Boothby, 2006, p. 253). Honwana's findings caution the potential harm that can be caused by transplanting trauma treatments across contexts and further indicate the potential benefit of utilising locally devised support options for survivors of trauma. Understanding the differences in trauma experience, interpretation and treatment needs may be vital to effectively implement TIC in LMICs.

Additionally, cultural and contextual considerations are relevant to the organisational component of TIC frameworks and

implementation efforts. Specifically, the barriers, facilitators and requirements of shifting an organisation toward TIC may vary from HICs to LMICs due to differences in organisational operations. De Carvalho et al. (2021) note the heterogeneity of healthcare systems in LMICs, but highlight the distinctions between healthcare systems in LMICs compared to HICs. De Carvalho et al. (2021)) identify resource limitations, segmentation (i.e., the concurrence of multiple different systems to serve different subpopulations within a country) and conditional investments from international donors as organisational factors that uniquely affect the Global South compared to the Global North. The unique organisational structures of LMICs may present challenges to implementing interventions for organisational changes. For example, Khan et al. (2018) identify multiple power imbalances (e.g., technical expertise and financial control) that allow external donors to influence health policy in Cambodia and Pakistan. This influence of donors may result in changes to the health system that are based on priorities/perspectives of other states/institutions, regardless of their suitability for the local context. As TIC is an organisational approach to change, differences in organisational structures from HICs compared to LMICs (e.g., accountability to external funders regarding priorities or changes) may warrant specific consideration in TIC implementation.

Due to cultural variations in trauma experiences, trauma responses and organisational structures, there is a need to understand TIC implementation within different contexts. However, the exclusion of LMICs from previous reviews (i.e., Bargeman et al., 2022; Mahon, 2022; Berring et al., 2024; Mahon, 2024) limits our ability to understand how TIC can be applied in different contexts. Rather than reflecting a lack of TIC implementation outside of HICs, findings may reflect the search strategies used. All four reviews utilised search strategies that centred on words related to TIC (e.g., "trauma-informed"). The creation and popularity of the term TIC is relatively recent. Therefore, there may be implementation efforts that are aligned with TIC but have been published without this term and thus excluded from previous reviews.

The current scoping review aims to create a more comprehensive understanding of organisational changes in LMICs that align with the principles of TIC, but which may not use this terminology. To do this, the review utilises a novel approach compared to previous TIC reviews. While previous reviews relied on terms specific to TIC (e.g., Bargeman et al., 2022), the current review will identify interventions that align with the organisational structure and principles of TIC to identify implementation efforts congruent with TIC regardless of their use of TIC terms. The review will then quantitatively capture (i) settings for TIC implementation, both geographically and by organisation type, (ii) the involvement of international organisations in implementation and (iii) the alignment of intervention aims to TIC principle(s).

## Methods

As the research question aims to map evidence from a developing field utilising diverse and interdisciplinary literature, a scoping review methodology was deemed most appropriate (Peters et al., 2020). The methodology was designed based on the Preferred Reporting Items for Systematic Reviews and Meta-Analyses-Scoping Review (PRISMA-ScR) Checklist (Tricco et al., 2018). The steps included registering the review, designing the research question, conducting the literature search, selecting studies, data extraction and data synthesis. The current review was registered on

Open Science Framework and updated throughout the review process (https://doi.org/10.17605/OSF.IO/EFCR2).

### Designing the research question

In line with best practices for scoping reviews, the research question was designed based on the constructs of context and concept (Peters et al., 2022). In the current review, context was defined as LMICs based on an adapted version of the World Bank's 2023 list (see Supplementary File 1 for the full list of countries; World Bank, 2024b). The adapted list includes countries that have been categorised as LMICs for the majority of the search period to account for countries with changing categorisations (e.g., Seychelles).

The concept of the current review is TIC. Based on SAMHSA's (2014) definition, TIC was conceptualised as involving two components: (i) an organisational change and (ii) an alignment to core principles. To capture organisational change, quality improvement (QI) literature was referenced. QI has been defined as a "focus on improving a process using 'the practice of continuously assessing and adjusting performance using statistically and scientifically accepted procedures' (College of American Pathologists, 2005, p.4) that will improve the quality of outcomes" (Compas et al., 2008). The second component of TIC was captured through the six core principles of TIC as defined by SAMHSA (2014): (1) safety; (2) trustworthiness and transparency; (3) peer support; (4) collaboration and mutuality; (5) empowerment, voice and choice; and (6) cultural, historical and gender issues. In combination with the first principle, the research design included organisational change that aimed to improve an organisation's design or operation with a goal of fostering the TIC principle(s) to support service users and/or providers.

### Literature search

The literature search was designed to capture the research question's context and concept as defined above. This included the World Bank list for LMICs with adaptations for countries with recent changes in their categorisation (as described above), an adapted search string from a QI review conducted by Compas et al. (2008)) and a string including common terms for TIC and the six SAMHSA TIC principles. The literature search was limited to research published in English (due to limitations in the review team's language fluency) and published from January 2000 until December 2023 to ensure the most current research was included. The search utilised the following search strings at the title, abstract and keywords level: (1) names of LMICs (2) terms related to organisational change including "Organizational change*" OR "Organizational innovation*" OR "Quality improvement*" (3) terms related to TIC including "Trauma informed" OR "Trauma awar*" OR "trauma Sensitive*" OR "TIC principle." A sample search string is available in Supplementary File 2.

The literature search was carried out across four databases on December 6 and 7, 2023. The search led to 1,940 results from Web of Science, 1,131 results from MEDLINE, 561 results from PsycINFO and 171 results from Embase. Following the screening processes, the first author (NM) reviewed the references of all included articles for any potentially relevant articles, leading to 206 articles being added for screening.

### Selecting studies

Results from the literature search were uploaded to Covidence, a web-based software that allows for reference management, screening and extraction by multiple user profiles. In total, 27 duplicates

were identified manually, and 890 duplicates were identified by Covidence. Following the removal of duplicates and including the results from the reference check, 3,092 articles were included for screening.

To facilitate the screening process, the inclusion and exclusion criteria were developed in collaboration with the research team, as shown below (Table 1).

To ensure reliability and accuracy between screeners, it was vital to consistently define and apply each of the TIC principles. To do this, the SAMHSA (2014) definition of each of the principles, in addition to examples of their application to the current review, was provided to all authors. An example of this is provided in Table 2 and the full table is included in Supplementary File 3.

SAMHSA's sixth principle aligns with concepts of intersectionality by focusing on cultural, historical and gender issues. This principle required further discussion among the review team to translate the concept into concrete criteria. Through discussion, the review team identified interventions aimed at reducing discrimination based on gender, mental health and race and ethnicity. With regard to reducing gender discrimination, it was decided to include efforts aimed at improving maternity care, including implementing respectful maternity care and reducing maternal mortality. This was based on literature demonstrating that high rates of maternal mortality are underpinned by gender discrimination within the medical field (Chirowa et al., 2013). Additionally, interventions aimed at including or supporting underserved communities, as defined by the studies' authors, were included. Finally, interventions aimed at supporting individuals with HIV+/AIDS were included. This inclusion was based on an understanding of the overlap between HIV and the principles of intersectionality. Namely, HIV care has been largely affected and limited by discrimination. For example, Beyrer et al. (2016)) outline the direct role stigma and discrimination against the LGBTQ+ community, and specifically men who have sex with men, have on HIV prevention and response efforts globally. Danil (2021) demonstrates the link

**Table 2.** Sample TIC principle definition

| Principle | SAMHSA definition | Inclusion/exclusion examples |
|---|---|---|
| Safety | "Throughout the organisation, staff and the people they serve, whether children or adults, feel physically and psychologically safe; the physical setting is safe and interpersonal interactions promote a sense of safety. Understanding safety as defined by those served is a high priority." | *Inclusion*: Practices to improve the workplace for staff, increase client comfort/safe access to services, support client and staff interactions *Exclusion*: Improvements to medical procedures (e.g., surgical checklist and safe medicine/anaesthesia administration processes), implementation to decrease negative medical experiences (e.g., infection rates and staff errors) |

between gender, racial and sexuality discrimination and colonisation in sub-Saharan Africa and connects this to modern challenges in effective and accessible HIV response. This definition of SAMHSA's sixth principle was discussed with all authors to ensure consistency within this review.

All titles and abstracts were screened by the first author (NM) and a second author. Conflicts in screening decisions were solved through discussion and consensus between the screeners. The title and abstract screening resulted in 499 studies to be screened at full-text. Full-text screenings were similarly completed by the first author (NM) and a second author (MWM, MÓS, TNA), with conflicts in screening decisions or exclusion reasons being resolved through discussion and consensus. Following the full-text screening, 255 were accepted for inclusion in this review. The references for all included articles are provided in Supplementary File 4. For the full selection process, see the PRISMA diagram below (Figure 1).

**Table 1.** Inclusion and exclusion criteria

| Construct | Inclusion | Exclusion |
|---|---|---|
| Language | Published in English | Published in a language outside of English |
| Publication | Peer-reviewed article | Non-peer-reviewed article (e.g., thesis, book, poster and conference abstracts, where a full-text, peer-reviewed article is not available) |
| Publication date | Published from 2000 onwards | Published before 2000 |
| Data | Primary data collection and analysis are conducted with a clear description of source data | A secondary data collection or analysis (e.g., meta-analysis and literature review) or a lack of description of source data (e.g., a description of program barriers without explanation of data source) or a lack of data (e.g., intervention description and study protocol) |
| Context | Implementation occurred in an organisation in a low- or middle-income country, based on the World Bank classification and specified for this study (see Supplementary File 1) | Implementation does not occur in an organisation in a low- or middle-income country as described in this study, or implementation occurs in multiple contexts, including a country beyond this study, and where data cannot be disaggregated by country |
| Construct- QI | A description of a Quality Improvement or similar initiative is provided where an existing organisation implements a change to their processes or service delivery | No description of an organisational change is provided, or the change implemented is not an organisational change (e.g., a single-day training of service providers) |
| Construct- TIC | The aim of the implementation is aligned with one of the six core TIC principles (i.e., safety; trustworthiness and transparency; collaboration and mutuality; empowerment, voice and choice; peer support; cultural, historical and gender issues) to create a better environment for service users or providers | The aim of the implementation is not aligned with one of the six core TIC principles nor is not to create a better environment for service users or providers (e.g., increase efficiency) |

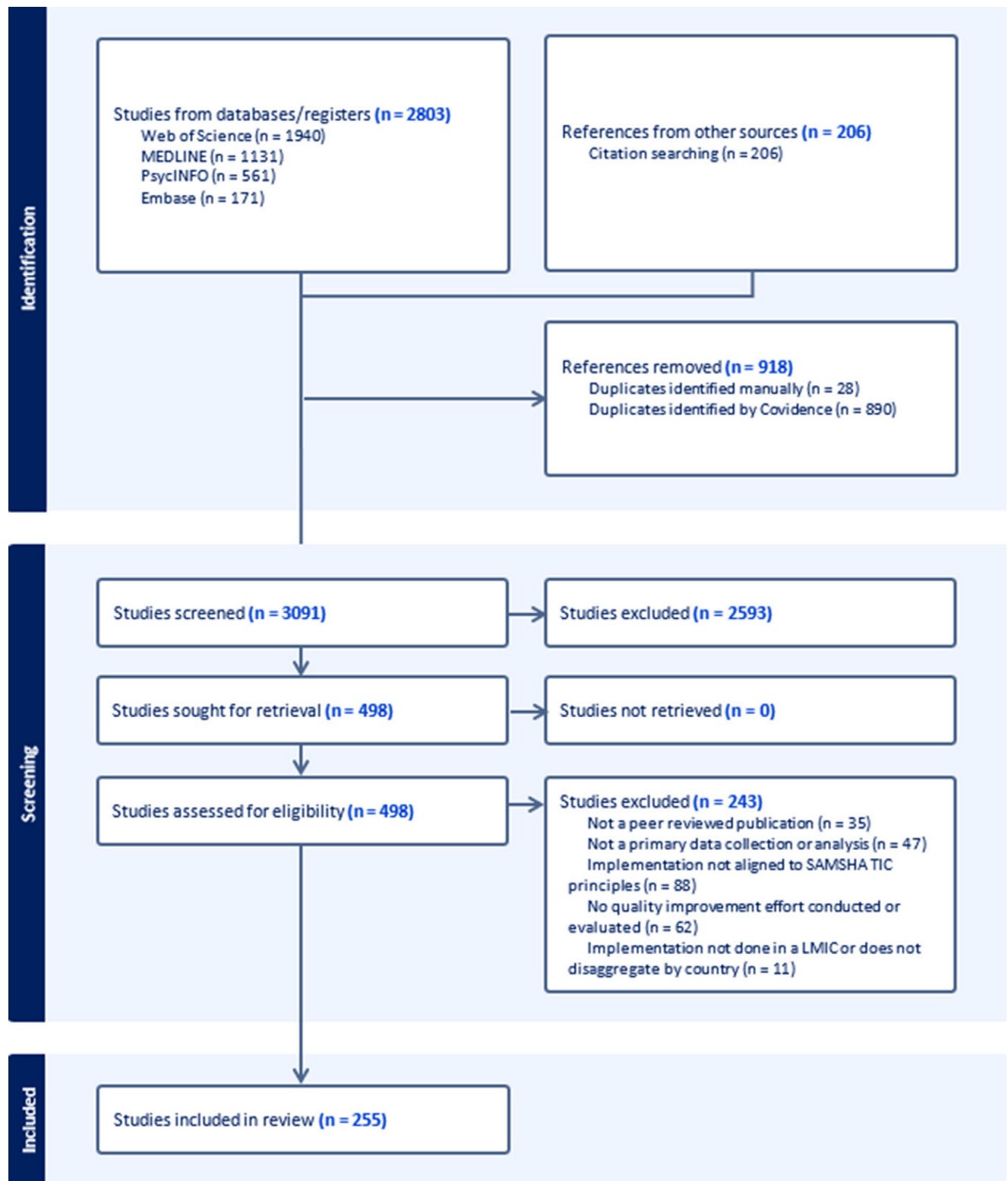

**Figure 1.** PRISMA flow chart.

### Data extraction

An extraction template was created in collaboration between the research team and uploaded to Covidence for extraction. The extraction template included the country of implementation, the names and countries of foreign funding and collaborating institutions involved in the study, the type of organisational setting (e.g., medical and education) and the SAMHSA principle (see Supplementary File 5 for full extraction template). Collaborating institutions included all foreign organisations with any role in the implementation (e.g., preparation, execution, evaluation and dissemination, including authorship on the study).

To support consistency in extraction, the first author (NM) simultaneously completed a portion of extractions with each of the authors participating in the extraction phase (MÓS, TNA, RC, CD, DSK and CS). The first author then met with each author to review extractions and reach a consensus on any conflicts in extracting. In total, 28 of the studies (10.94%) were extracted by two authors. The remaining studies were extracted by one author. The extracted data are available in Supplementary File 6.

### Data synthesis

Following extraction, data were uploaded to Statistical Package for Social Sciences version 28. Foreign funding and collaborating institutions were clustered based on broad descriptions of the type of organisation defined in Table 3.

Descriptive statistics were conducted to analyse the frequency of the extracted data. Extracted and coded data are provided in Supplementary File 6.

### Results

The purpose of the current review was to understand the occurrence of interventions that align with the constructs of organisational change and principles of TIC in LMICs. A description of these efforts is included below. Further details of each finding are provided in Supplementary File 7.

Of the 255 included articles, 24 were implemented in multiple countries and 231 were implemented in one of 39 LMICs. Implementation efforts spanned five world regions as defined by the World Bank (2025). The majority of studies were conducted in the Sub-Saharan region ($n = 147$; 63.64%). The single country with the most studies was India ($n = 29$; 11.4%), followed by South Africa ($n = 21$; 9.09%).

In addition to analysing the country of implementation, the involvement of other countries was analysed. The role of other countries was divided into two categories: funder or collaborator (i.e., any role in paper authorship or the design, implementation, evaluation or dissemination of the study). On 11 occasions, the involvement of other countries was identified without specification of the role of the country (e.g., identified a country as providing "support") and could not be included in this analysis. Only 23 (9%)

**Table 3.** Type of funding/collaborating organisation

| Organisation category |
|---|
| Academic institution (e.g., university and college) |
| Business/corporation (e.g., consulting firm) |
| Medical institution (e.g., hospital and clinic) |
| Multi-country governmental organisation (e.g., WHO, UN and EU) |
| National governmental organisation/department/public body (e.g., agency and department) |
| Non-governmental organisation/non-profit/not-for-profit/charity |
| Private foundation (e.g., philanthropic organisation) |
| Professional body (e.g., profession-based society) |
| Research centre/institute (e.g., research organisation) |
| Other |

of studies were conducted without the involvement of foreign institutions.

At least one funder was reported by 184 (72.16%) of the studies, with 304 funding streams being reported across the studies. The United States was the most frequent funder country, accounting for 179 (58.88%) of the funders. The most common funding organisation type was national governmental organisations, departments and public bodies, which accounted for 128 (42.11%) of the funders reported.

The involvement of foreign institutions through participation in the research was similarly analysed. Two hundred and eight (81.6%) studies reported the involvement of foreign institutions, totalling 588 institutional collaborations. The most common country of these institutions was the United States, which encompassed 327 (55.61%) of the institutions, followed by the United Kingdom, which encompassed 59 (10.03%) of the institutions. The most common collaborating institution was academic institutions, accounting for half of the collaborations.

To understand TIC efforts, the types of organisations implementing TIC within the included studies were analysed. Two hundred and thirty-one (90.6%) studies were implemented within medical settings such as hospitals or clinics. The remaining implementation efforts took place within mental health settings, educational settings, governmental departments or organisations, across multiple settings or in other settings outside of these categories (e.g., police force and workplace).

Finally, an analysis of the principle(s) that implementation efforts aimed to address was conducted. The majority of efforts aimed to address the impact of cultural, historical and gender issues (176 studies, 69.02%). Forty-seven (18.43%) interventions aimed to address multiple TIC principles. Of the efforts aimed at addressing cultural, historical and gender issues, 120 studies (68.18% of studies included in cultural, historical and gender issues) involved efforts related to gender. This included efforts such as increasing respectful maternity care, improving access to contraception and supporting maternal health. Forty-nine studies (27.84% of studies included in cultural, historical and gender issues) involved efforts related to HIV+ care. This included efforts related to access to HIV testing and antiretroviral treatment and HIV support services. The third most common effort within cultural, historical and gender issues aimed to address multiple components of this principle and included 28 studies (15.91% of studies included in cultural, historical and gender issues). Studies aiming to address multiple components of this principle included efforts to address gender and HIV+ equity (e.g., improving maternity care for women with HIV+ status), efforts to address gender and LGBTQIA+ equity (e.g., addressing gender and sexuality inequality in sexual and reproductive health projects) and efforts to address gender and underserved populations (e.g., improved maternity care in a region of low socioeconomic status). Figure 2 shows the principles of focus within the included studies.

### Discussion

The current review indicates that interventions aligning with the constructs of TIC have been implemented in and published by organisations in LMICs. The literature represents diverse interventions, including a range of implementation contexts, collaborating countries and the application of TIC principles. This finding expands on previous reviews (e.g., Mahon, 2022) by considering interventions that are published without specific use of terms like

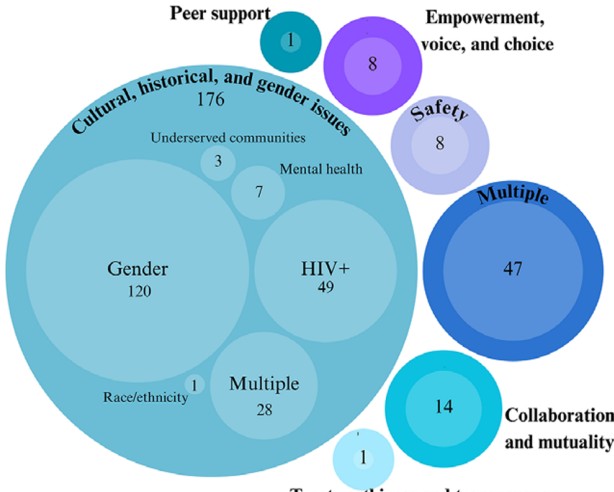

**Figure 2.** TIC implementation principles.

"trauma-informed" and has direct implications for future interventions and research. Although the current results capture a diverse range of literature, of note is the high prevalence of medical settings as implementation sites, indicating that interventions that align with TIC in LMICs are more likely to be implemented within medical settings. Substantial literature has documented concerns about the state of healthcare in LMICs and its impacts (Kruk et al., 2018). Thus, the current results may reflect research priorities and an investment in improved medical care in LMICs. However, the findings could also be affected by skew in publication. Specifically, research demonstrates that humanities and social science scholars publish less frequently in scientific journals due to a preference for a wider range of publication formats (e.g., book chapters) and are more likely to publish in languages outside of English (Nederhof, 2006; Pajić et al., 2019). The authors' choices to exclude other forms of literature (e.g., book chapters and grey literature) and research published outside of English may have skewed the current results towards implementation efforts based in medical settings. Regardless of the potential effect of skew in publication, the results suggest that TIC principles may be well-suited to medical settings. Future research can analyse the efficacy of these interventions with particular attention to contributing factors that may support or hinder their implementation (e.g., resources and approaches). An analysis of these factors may help to inform efforts in other settings.

The included interventions showed geographic diversity in implementation countries spanning 24 LMICs. Approximately 20% of the interventions within the review were implemented in two countries, namely India and South Africa, respectively accounting for 11.4% and 9.09% of the implementation countries. The high frequency of interventions in medical settings and interventions aimed at cultural, historical and gender issues may be particularly relevant for these two contexts. Specifically, the South African medical system was fragmented and under-resourced during apartheid, creating an inequitable system, particularly for Black and low-income South Africans, with ripple effects to the modern-day system (Baker, 2010). India's healthcare system faces similar equity challenges with health inequity occurring based on social classes, such as income, gender, caste and geography (Balarajan et al., 2011). While the frequency of these countries in the current findings could be interpreted as a reflection of these challenges, it could also be a reflection of the emphasis on improvement of healthcare

systems. Within South Africa, health care is a constitutional right that has been advanced through multiple government actions (Maphumulo and Bhengu, 2019). India has also seen significant growth in its medical system and actions to support the quality of care (e.g., increases in accreditation systems; Rudrappa et al., 2019). Therefore, the current findings may represent political will that supports and prioritises improvements within India and South Africa.

Beyond implementation settings, the current review indicates that the vast majority of interventions aligned with TIC principles were aimed at improving cultural, historical and gender issues – specifically, gender issues and HIV care. Concerns related to gender issues – in particular, maternal mortality and contraception action and HIV prevention and response programming – are matters of high priority within global health research agendas (UN DESA, 2024). The prevalence of these challenges, high mortality rates and preventability of these issues underlie efforts to reduce their occurrence, particularly within LMICs (Fettig et al., 2014; Ekwuazi et al., 2023). The frequency of these efforts in the current review aligns with findings related to implementation countries. The high occurrence of interventions in Sub-Saharan Africa, specifically South Africa , may be related to the high frequency of HIV programming given the prevalence of HIV in this region (Dwyer-Lindgren et al., 2019). Similarly, India's high maternal mortality rate has led to large-scale efforts and investments in improving maternity care, which have demonstrated success (Ministry of Health and Family Welfare, 2025). Thus, the results related to implementation countries and priorities may reflect sustained prioritisation of issues with high prevalence and mortality in implementing countries.

In considering the aims and priorities of the implementation efforts captured in the current review, findings related to foreign involvement need to be considered. The involvement of foreign funders and collaborators in the vast majority of studies, approximately half of which are from the United States, raises questions about research agendas. LMICs have traditionally been excluded from setting research agendas (Charani et al., 2022). Furthermore, global health's roots in colonialism have directly led to power differentials between HICs and LMICs maintained by neocolonialist systems – for example, modern funding structures (Kwete et al., 2022).

Considerations for modern funding structures are of particular importance in relation to the frequency of US funding identified within this review due to the current sociopolitical climate within the United States. The multi-billion-dollar cuts to foreign aid and dismantling of the US Agency for International Development (USAID) under President Donald J. Trump pose immense health risks to LMICs. Estimates of mortality from AIDS alone are predicted to rise by an additional 15 million people by 2040 due to cuts (Mallapaty, 2025). Neocolonial approaches to funding, such as USAID, have led to LMICs' forced dependence on HICs' goodwill, causing precarious funding environments that limit the local production of TIC to that of Westerners' conceptualisation(s). In the face of cuts to these programs, the sustainability of interventions aligned with the principles of TIC is being challenged. While cuts to funding will impact future TIC-aligned interventions, it is unclear whether the interventions captured in this review were a result of locally identified priorities and methods or whether they were superimposed by external funders and collaborators. To make this connection clear, future research should analyse the power structures in TIC-aligned interventions, and implementors should make explicit the methods of their

conceptualisation and operationalisation of TIC. In alignment with the TIC principle of historical, cultural and gender issues, future TIC-aligned interventions should highlight the legacies of historic oppression, imperialism and colonisation that disproportionately affect LMICs.

This review indicates that multiple implementation efforts with the same aims as TIC are conducted without being named as such. Theoretically, this raises questions about defining TIC. These questions are not new, as previous reviews indicate that TIC research utilises poorly defined models of TIC (Bargeman et al., 2022; Mahon, 2022) and varies in its definition and application of TIC principles (Berring et al., 2024). Further, although TIC was coined by Harris and Fallot in Harris and Fallot, 2001, multiple other terms, advancements and approaches overlap with this concept. For example, Ekman et al. (2011) compare the terms "patient-centred," "personalised medicine" and "person-centred," all of which overlap with one or more TIC principles, such as voice and choice. Similarly, recovery-oriented care (ROC), a model also largely developed by SAMHSA with a focus on mental health and substance use, demonstrates similarities to TIC. Both TIC and ROC utilise a systems approach considering multiple organisational components (e.g., service delivery and financing) and contain overlapping principles (e.g., empowerment, peer-support and recognition of cultural factors; Davidson et al., 2021; Sheedy and Whitter, 2013). Based on this literature and the current results, there is a need for researchers to place TIC within Global research, explicitly identifying its convergence and divergence from other intervention approaches.

Beyond theoretical contributions, the current review can inform TIC implementation efforts within LMICs. As TIC increases in popularity, there may be an increase in efforts to deliver TIC as a new model in LMICs. While there exists a dearth of ongoing TIC research in LMICs (e.g., Bargeman et al., 2022), the current results suggest that interventions aligned with TIC are currently being implemented. These interventions require consideration in future TIC interventions to avoid a repetition of efforts. Before implementing TIC interventions, an organisation should conduct a comprehensive evaluation to fully capture practices that align with the TIC framework, regardless of their use of TIC terms. The aim of this evaluation should be to identify current trauma-informed practices that can be reinforced and supported, and to identify practices divergent from TIC principles that should be addressed in interventions. Through this assessment, organisations can create specific interventions based on their unique organisational needs and contexts.

### Limitations and future directions

Multiple limitations may have affected the current results, possibly reducing the number of articles included within the current review. First, as described above, the review was designed to focus on primary data collection and analyses published within scientific peer-reviewed journals. The authors planned to include a grey literature search of reports from governmental websites and non-profit organisations; however, due to the size of the peer-reviewed literature, this was not deemed feasible. With 255 articles included in the current review, the feasibility of incorporating diverse grey literature into a coherent narrative with the existing vast publications raised concerns.

Second, the use of QI literature as a core component of the review's construct may have limited the review results. It is possible that interventions that include organisational change in alignment with QI were published without the use of such terms, preventing their inclusion in this review. This may also, at least partially, account for the focus on medical settings within the current review, as QI has been extensively used within medical research, particularly in LMICs (Odhus et al., 2024). Therefore, the search strategy may have been biased towards medical settings.

Third, interventions were only included if their aim was aligned with one of the TIC principles, rather than their methods. For example, a study by Alidina et al. (2022)) implemented a mentorship program with the aim of increasing the quality of surgical care. Although the mentorship approach as a method may align with the TIC principles of peer support or collaboration and mutuality, this study was excluded as the objective of increasing surgical care is not aligned with TIC. Collectively, these limitations suggest that there may be a greater amount of literature that aligns with TIC if grey literature, broader search terms and the method, in addition to the objective, were considered in the current review.

In addition to limitations in the review's search and screening strategy, the current analysis is limited to quantitatively describing the context and aims of identified interventions. The choice to focus on quantitative results was based on the size and heterogeneity of results as described above. However, future research may benefit from further qualitative analysis of interventions that align with principles of TIC. To create a clear and coherent narrative, researchers undertaking this endeavour should narrow their reviews based on the elements of interventions that are most relevant to their work (e.g., TIC principle, organisational context and country of implementation). By creating a more focused review specific to a context or TIC principle, researchers will be able to conduct analyses with more depth. This may include a description of intervention strategies, barriers and facilitators to interventions and interactions between variables identified in the current review, such as intervention contexts, foreign collaborators/funders and intervention aims.

Despite limitations, the current review captured a wide breadth of diverse interventions that align with TIC and were conducted in LMICs. Due to the heterogeneity of the field and TIC implementation, it may be impossible to capture all these efforts in one review. Rather, the current review should serve as an indicator that TIC is happening in diverse settings and under diverse names.

Given the variation in trauma experiences and responses (Patel and Hall, 2021) and organisations (De Carvalho G et al., 2021) across contexts, future research should aim to develop TIC implementation tools that support its use in diverse settings. This research should be based on priority-setting efforts within LMICs to identify organisational contexts that could most benefit from its use. Additionally, efforts should include locally defining TIC principles with individuals within those organisations and transparently disseminating this process. TIC as a framework may be particularly suited for this, as demonstrated in this review. For example, the inclusion of HIV in the current review based on the connection between the legacy of colonisation and modern HIV care across the global South (Danil, 2021) is a testament to the adaptability of this framework. Specifically, the sixth SAMHSA principle makes TIC particularly well suited to local definition and contextualisation, supporting its global use; the weight of doing this meaningfully and tangibly, though, rests on those who hold the power in implementation efforts. Future TIC research and resulting theoretical and implementation advancements (e.g., toolkits) need to be clear in who develops TIC, and with whom, to avoid the continuation of a TIC model based on a minority of the global population.

## Conclusions

The current review expands on existing understandings of TIC implementation by capturing literature that aligns with the structure and principles of TIC without necessarily using the term TIC. In doing so, the review identifies intervention efforts across LMICs that include the core components of TIC. The methods and results of this review raise larger questions for the development of TIC as a theory and practice. Specifically, the current review suggests that future theoretical conceptualisations and practical applications of TIC should consider intersecting work that is not labelled as "TIC." As TIC develops, researchers and implementers will need to both explicitly place their work within the wider trauma field and locally define their models. Implementation efforts in LMICs that include foreign funders/collaborators require explicit descriptions of the role(s) of foreign bodies in their work. Further, for TIC to be localised, foreign funders/collaborators should make clear how they accounted for existing power imbalances. TIC should ideally be co-produced at the local level, led by local stakeholders with organisational knowledge and expertise, and designed for and by the service users and providers it affects most.

**Open peer review.** To view the open peer review materials for this article, please visit http://doi.org/10.1017/gmh.2025.10111.

**Supplementary material.** The supplementary material for this article can be found at http://doi.org/10.1017/gmh.2025.10111.

**Data availability statement.** The authors confirm that the data supporting the findings of this study are available within the Supplementary Materials of this article.

**Author contribution.** NM: Conceptualisation, data curation, analysis, funding acquisition, investigation, methodology, project administration, validation, visualisation, writing – original draft and writing – review and editing. MR: Conceptualisation, funding acquisition, methodology, supervision, and writing – review and editing. MWM, RC, CD, DSK and CS: Investigation. GS: Conceptualisation and methodology. MS and TNA: Investigation and writing – review and editing.

**Financial support.** NM received funding from the Trinity Research Doctorate Award at Trinity College Dublin, Ireland. MWM received funding from the Arts and Social Sciences Benefactions Fund at Trinity College Dublin, Ireland.

**Competing interests.** The authors declare none.

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
