## [Reviewer Report]

1. General

a) This is a methodologically sound review addressing an important and underexplored topic. The contribution is both original and meaningful. I only have minor suggestions for improvement.

2. Introduction

a) Page 4, lines 173–175: You note that your review captures implementation efforts consistent with TIC principles, even when the term “trauma-informed care” is not explicitly used. This is a significant and novel strength. I recommend briefly introducing this earlier in the introduction—perhaps by explaining that many such interventions may have been excluded in previous reviews—to underscore the importance of your work.

3. Methods

a) Table 1: I suggest listing the TIC principles either within Table 1 or in the main text that describes the inclusion and exclusion criteria, for the sake of transparency and reproducibility.

4. Results

a) Page 15: The sentence “implemented in one of 39 countries across 5 of regions defined by the World Bank (2025)” could be clarified as: “implemented in one of 39 countries defined by the World Bank (2025) as low- and middle-income countries (LMICs).”

b) Both in the introduction and discussion, you mention the risks related to foreign support and HIC-LMIC power dynamics in shaping priorities for organisational change. Please consider analyzing your extracted data to assess whether the type of organization and the involvement of international stakeholders influenced alignment with specific TIC principles. This would substantially strengthen your discussion on global power dynamics.

c) In the discussion section, you state: “This review indicates that multiple implementation efforts with the same aims as TIC are conducted without being named as such.” Please consider reporting in the Results section how many studies met TIC principles without explicitly referring to them, and detail which principles were most often aligned with. Additionally, please indicate whether these interventions were framed under alternative models, such as recovery-oriented care (ROC) or patient-centred care.

5. Discussion

a) Page 17, lines 367–370: You mention other frameworks overlapping with TIC. Please consider discussing the theoretical overlap between recovery-oriented care (ROC) and TIC. For example, ROC emphasizes principles such as safety, peer support, and empowerment—core elements of TIC—and includes interventions like Safewards and the Six Core Strategies, which refer to both trauma-awareness and recovery. The WHO QualityRights initiative, particularly relevant in LMIC contexts, also explicitly calls for trauma-informed approaches within its ROC framework. I suggest elaborating on this convergence between ROC and TIC.

6. Conclusion

a) Page 19, lines 425–434: Your conclusion is clear and well-aligned with the aims of the review. To strengthen your final message, consider ending with a call for increased coproduction of organizational change efforts and active involvement of local stakeholders at the management level to guide future implementation initiatives.

---

## [Reviewer Report]

Question 1:

For global reviews, how well does the review cover global content in the inclusion of

research, presentation of results, and/or in the discussion and implications? And how could this be improved/expanded?

- The authors focus their scoping review search on studies from low and middle income countries (LMICs). They thoroughly define how LMICs are defined. Some feedback is provided (see point 7) regarding how their definition can be further clarified.

Question 2:

For reviews that are regionally focused, how well do the authors describe how the results fit in with global research and global learnings? And how could this be improved/expanded?

- The current scoping review was not regionally focused. However, the authors describe the importance of applying global findings in a way that is contextually and locally relevant.

Feedback

Thank you for the opportunity to review the manuscript, "Trauma informed care (TIC) in low and middle income countries: A scoping review of organisational implementation efforts. The authors have conducted an important review to fills a gap in our global understanding of TIC. They creatively expanded their search by being broadly inclusive of search terms so as to not inadvertently exclude studies focused on TIC in LMICs. A scoping review was a suitable approach for their study aims given the limited research in this area. There are several strengths to highlight. The authors are to be commended for identifying and screening through such a large number of articles. This project was a large undertaking! The authors provide a good rationale for how the research question was designed. Similarly, their data extraction method is clearly articulated. Below, I provide recommendations for further strengthening the manuscript. Some are minor edits but points 8 and 9 can be considered major revisions. In particular, given the large number of articles that the authors identified, I was expecting to see further elaboration on the differing definitions of TIC and/or how are they are implemented in LMIC contexts. There is mention of how TIC work is happening but further details are not provided. For example, what does TIC work look like across contexts? What are similarities? What are differences? Are there challenges with implementation? Lessons learned for the Global North? These questions would be best answered by qualitative analysis of the identified articles (in addition to the quant analysis presented). It’s possible these questions cannot be answered in this scoping review but I would recommend articulating the parameters and “right-sizing” the goals of the scoping review in the Intro if that’s the case. If a qualitative analysis is beyond the scope of this manuscript, I would encourage the authors to consider a separate manuscript where qualitative data is extracted from the identified articles as there can be much to learn from synthesizing that data. Further feedback is provided below.

1) Suggest including more recent prevalence rates based on DSM-5 or recent ICD definitions (in the Introduction).

2) Grammar suggestion - check that ‘while’ and ‘although’ are used appropriately. “While” typically connotes the passage of time; “although” is for comparison.

3) Please do a thorough proofread. Some errors noted below but check throughout please.

4) Line 163 - sentence is confusingly worded.

5) Line 179 - “developing field” instead of “developing fielding”

6) Line 183 - “designing the research question” mentioned twice

7) Line 209 - can you further clarify what the adapted World Bank list for LMICs is? Is it adapted because it includes countries with changing status as described in the ‘Designing the Research Question’ section? Or are there are reasons it is described as adapted?

8) Further elaboration in the Results and Discussion are needed. For instance, (line 350) indicates high involvement from the US. What are the implications of this for sustainability in the current climate when US funding has been significantly cut? The considerations from a lens of historic oppression, imperialism, and colonisation are considered but it’s important to extend the link to the current climate.

Relatedly, only quantitative information is provided in the Results (e.g., number of funders/funding streams reported, involvement of other countries...etc). However, further qualitative analysis of other key findings would strengthen and deepen the results. I appreciate that is a bigger undertaking. If the authors deem that is beyond the goals of this scoping review, I would recommend setting the stage in the Introduction that the current scoping review solely focused on quantitative data from the studies. I would also strongly encourage the authors to either integrate qualitative themes in the current scoping review or consider a second manuscript with the qualitative themes can be elaborated on. This also relates to point number 9 below.

9) Please elaborate further on the differing TIC definitions. Are there key components of the definition that are important to consider as new definitions are explored? What is still similar to existing definitions based on SAMHSA principles?

10) Line 382-383 - change language to active voice and check subjects in sentence structure. Instead of “the review chose to”, change to “the review was designed to focus on primary data collection...” OR “the authors focused on identifying studies based on primary data collection”; “the authors planned to include a grey literature search...”. This approach indicates that it is the authors' decision; the review itself can’t make these decisions.

---

## [Reviewer Report]

Thank you for addressing most of the suggestions or better acknowledging the scope and limitations of your work. Thank you for your contribution to this field.

---

## [Reviewer Report]

Dear Editors,

Thank you for the opportunity to review the revised manuscript, "Trauma informed care (TIC) in low and middle income countries: A scoping review of organisational implementation efforts”.

The authors have conducted an important review in efforts to fill a gap in the literature regarding global understanding of trauma-informed care.

Although the authors have responded to the feedback provided in the first round of reviews, I would recommend another major revision. My reason for this is because more depth is needed in the Results and Discussion sections to meet the bar for a published manuscript. I have included specific recommendations below for how the authors can elaborate on the Results and Discussion sections. The authors note that they reviewed 255 articles, which I acknowledge is a massive undertaking in and of itself. Given the large number of articles included in the review, readers would benefit from learning more about the findings from this review beyond frequencies.

I understand the rationale for focusing on the quantitative results given the large number of articles to review. At the same time, the authors can still capture nuances in the quantitative findings (see below for suggestions). If space is a challenge, I would request that the Editors make an exception given the additional depth required to present the results from a scoping review.

Please see below for my feedback.

Line 99: Recommend citing DSM-III when you mention the introduction of PTSD (this is the primary source, rather than Andreasen’s work).

Line 155: “included” mentioned twice - seems like an error?

Line 163: change “forceable” to “forcible”

Line 169: “child soldiers” used more commonly than “children soldiers”

Line 172-176: “In a longitudinal study that followed child soldiers in Mozambique from 1988 to 2004 both the former child soldiers and members of their communities identified traditional ceremonies as supportive of the child soldiers’ reintegration to society. The ceremonies were “local beliefs put into practice, they facilitated the re-alignment of individual, family and communal relationships” (Boothby, 2006, p. 253)”

Wording is confusing. That being said, I think this can be removed as it’s not directly related to TIC

Alternatively, strengthen the link between this example and the implementation of TIC. What is the relationship between the integration of traditional ceremonies and TIC?

Line 190: Again, please provide a stronger connection between the example from Khan et al (2018) and TIC. For every example you cite, it’s helpful to “hold the reader’s hand” and describe how this information relates to TIC.

Line 216: the word “aim” seems misplaced. Can delete.

Line 254: Indicate the last day of the search (i.e., Dec 7th, 2023). As such, the lit search was limited from January (?) 2000 to December 2023

Line 266: “web-based” instead of “web-passed software”

Selecting Studies - PRISMA Flow chart is mentioned on line 309. Please move reference to Figure 1 earlier when you describe how studies were selected.

Relatedly, please add more details in the PRISMA diagram for ‘studies excluded’ (n=2593) in the Screening category. What is the difference between the studies excluded in the first box (i.e., 2593 studies) and those in the second box (n=243)?

Line 278: I believe this should be noted as “Table 2”, not “Table 3”

Line 286: “Within reducing gender discrimination” is awkwardly phrased. Perhaps change to “with regard to” (or something along those lines)

Line 291: Provide an example of what you mean by “contextualized definitions” (similar to the previous sentence where you elaborate on improving maternity care).

Line 296: Updated acronym for LGBTQ is LGBTQ+

Line 305: which second authors assisted with screening full texts? Initials only noted for first author.

Line 318: integrate the sentence, “The country and name of…” as part of the previous sentence that highlights what the extraction template included

Line 337: Change to “By conducting the current review, we aimed to understand…” (or something along these lines of a more active, rather than passive, voice). As noted before, the current review cannot understand something - only those conducting the review can. I understand the general approach to use third person when writing but there is movement towards writing in first person, which is acceptable in this case. If you prefer to keep third person, you can also change language to “the purpose of the current review was to…”

Line 341-345: Somewhat confusingly worded. Particularly, I’m not sure what this section of the sentence means: “were implemented in one of 39 LMICs, representing 5 of world regions as defined by the World Bank (2025). “

See suggested edits below (but please also edit the sentence noted above as I’m not sure what that is trying to convey):

Of the 255 included articles reviewed, 24 programs were implemented in multiple countries and 231 programs were implemented in one of 39 LMICs, representing 5 of world regions as defined by the World Bank (2025). The majority of studies were conducted in the Sub-Saharan region (n=147; 63.64%). The single country with the most studies was India (n=29; 11.4%) followed by South Africa (n=21; 9.09%).

Line 349: “another” and “other” used - delete one of these

Line 353: “At least one funder was reported by 184 (72.16%) of the studies, with 304 funding streams being reported across the studies” - Does this mean that the US was the funder? Or is there a specific funder within the US that was reported by the majority of countries?

Line 357: percentage sign missing for “42.11”

Line 359: remove the words “participation of”

Line 366: percentage used here but throughout, numbers are provided. Suggest including numbers here too (and in subsequent paragraphs, e.g., line 372) for consistency.

Results

Please include more substantial information here. Unfortunately, I don’t see much more depth added here since the last submission. As the Results stand, they don’t give the reader much information on the nuances of implementation efforts beyond the statistics of frequency etc.

Figure 2 is a nice visual. It could even be helpful to elaborate on this visual since qualitative findings are not included.

For example, line 369 - give examples of “other settings”

For example, lines 370-373 - you can elaborate much more here on what you mean by the ‘impact of cultural, historical and gender issues’. Please define these terms and describe a) what was being measured, and b) what are some nuances in the findings here?

NB: These are only examples so please feel free to go beyond the suggestions here.

Discussion - more depth here would be helpful. See below for specific feedback:

Thanks for including additional literature on related TIC models and highlighting the limitations of focusing on the quantitative results.

Thank you as well for elaborating on the current sociopolitical climate and impacts on funding. To make the link directly to your review, it would be helpful to make a recommendation for ongoing work to understand the implementation of TIC in LMICs, given the shifting landscape of funding options. For instance, future research can explore some of the following questions:

What is the impact of TIC implementation in the face of substantial funding cuts?

Are TIC efforts implemented sufficiently (and contextually grounded already) to be sustainable without funding?

Such an approach will emphasize the connection between your research, funding cuts, and future work.

Lines 427 - 434 - This seems a bit oddly placed. I’m not sure how the baseline evaluation recommendation connects to the broader results. Please make a stronger connection for how this information connects to your results. For instance, how can the findings from your review inform a baseline evaluation within a particular organization? If the link doesn’t seem strong enough, I would suggest removing this section.

Instead, it’s important to elaborate further on your Results in the Discussion section. What do the findings suggest? What are some nuances you can draw on from your findings? Unfortunately, I don’t see much of these details included in the Discussion section (beyond the brief mention of the majority of studies being conducted in medical settings).

What are some reasons why TIC is mainly seen in medical settings? You noted funding priorities but any other reasons? How might other settings benefit from integrating TIC? Are there lessons that can be learned from existing implementation efforts in healthcare settings?

Since India and South Africa were identified as the single countries with the most studies, consider expanding on some of the findings from the studies here. What are some interpretations you can draw from the findings across these two countries? What are some reasons that TIC efforts are higher here (or at least there is more published literature to indicate as such).

Line 417 - “TIC principles” instead of “TIC principle”

Line 477 - “given the variation in trauma” - reword this. Do you mean, “given the variation in definitions/conceptualizations of trauma” ?

Your Conclusion section makes valuable points. Your discussion section can be strengthened by elaborating on these points further. For instance, did any of the papers you reviewed describe power imbalances? Was there discussion about how / if TIC was co-produced? You don’t have to do a systematic qualitative review of such questions but your discussion section can be stronger by drawing on such examples.